# Spontaneous dormancy protects *Trypanosoma cruzi* during extended drug exposure

Fernando J Sánchez-Valdéz[1†‡], Angel Padilla[1,2†], Wei Wang[1], Dylan Orr[1], Rick L Tarleton[1,2*]

[1]Center for Tropical and Emerging Global Diseases, University of Georgia, Athens, United States; [2]Department of Cellular Biology, University of Georgia, Athens, United States

**Abstract** The ability of the Chagas disease agent *Trypanosoma cruzi* to resist extended in vivo exposure to highly effective trypanocidal compounds prompted us to explore the potential for dormancy and its contribution to failed drug treatments in this infection. We document the development of non-proliferating intracellular amastigotes in vivo and in vitro in the absence of drug treatment. Non-proliferative amastigotes ultimately converted to trypomastigotes and established infections in new host cells. Most significantly, dormant amastigotes were uniquely resistant to extended drug treatment in vivo and in vitro and could re-establish a flourishing infection after as many as 30 days of drug exposure. These results demonstrate a dormancy state in *T. cruzi* that accounts for the failure of highly cytotoxic compounds to completely resolve the infection. The ability of *T. cruzi* to establish dormancy throws into question current methods for identifying curative drugs but also suggests alternative therapeutic approaches.

DOI: https://doi.org/10.7554/eLife.34039.001

*For correspondence:
tarleton@uga.edu

†These authors contributed equally to this work

Present address: ‡Instituto de Patología Experimental-CONICET, Universidad Nacional de Salta, Salta, Argentina

Competing interests: The authors declare that no competing interests exist.

## Introduction

Parasitic diseases have been particularly resistant to the development of effective vaccines and safe and curative therapeutic drugs. The drugs that have been put into use often suffer from one or more of the problems of inefficiency, high toxicity and/or selection of drug resistant mutants.

Despite being one of the highest impact infectious diseases in the Americas, *T. cruzi* infection in humans is relatively rarely treated. Initially this was because of the misconception that Chagas disease had an autoimmune, rather than pathogen persistence etiology (*Tarleton, 2003*) but more recently the substantial toxic side effects and the limited supply of benznidazole (BZN) and nifurtimox (NFX), drugs that have been in use for decades, has also contributed significantly to their low rate of utilization. However, the highly variable outcome of treatment regimens employing these compounds – ranging from 0% to 100% depending on the study - and the difficulty of accurately assessing these outcomes, also keep utilization as low as 1% of those infected (*de Castro et al., 2006*; *Guedes et al., 2011*; *Jackson et al., 2010*; *Navarro et al., 2012*; *Yun et al., 2009*).

The reason for the high variability in the efficacy of BZN and NFX is not known but has been primarily attributed to the broad genetic diversity of *T. cruzi* isolates. Importantly, the lack of drug efficacy is almost certainly not a result of selection of genetic resistance by drug use or misuse, as neither BZN nor NFX have been widely or indiscriminately used and the induction of stable resistance to these drugs is difficult to produce in vitro or in vivo. Further, the variable and often low contribution of humans to infection of insect vectors (many infections in insects originate from feeding on non-human mammals, within which *T. cruzi* circulates widely and at high levels [*Cohen and Gürtler, 2001*]), offers very few opportunities to drive and spread resistance even if it were to develop.

**eLife digest** Chagas disease is one of the most harmful tropical diseases in the Americas. It affects millions of people, predominantly in Latin America. It is usually spread by kissing bugs infected with *Trypanosoma cruzi* parasites. It is considered a neglected tropical disease because few effective treatments and preventive methods are routinely used.

Several drugs can kill *T. cruzi* parasites, but they often fail to cure the infection. Many people with Chagas disease go on to have life-long infections and eventually develop heart failure. The reason for the high rate of treatment failure is not known. It does not appear to be the result of the parasites developing resistance to the drugs. One possibility is that the parasites can hide in a dormant state in the body, dodging the toxic drugs and living to reproduce another day.

Now, Sánchez-Valdéz et al. identify a dormant form of the *T. cruzi* parasite that allows the infection to persist after treatment. In the experiments, a non-reproducing form of the so-called amastigote stage of the *T. cruzi* parasite inside the host cells was observed in infected mice and human cells. While some of the amastigote parasites continue multiplying, a few stop even without drug treatment – but can resume multiplication at a later time. They may also be able to change into the trypomastigote form of the parasite, which can infect new cells. These non-multiplying amastigotes can survive drug treatment for as long as 30 days, whereas the multiplying amastigotes are killed by such drugs. However, the surviving amastigotes then reestablish active infections after treatment has stopped.

The experiments explain why treatment so often fails to cure Chagas disease. This suggests new treatment strategies are needed, including using existing drugs for a longer time perhaps with less frequent doses. New therapies that kill the dormant amastigotes may also help. Treatments that overcome the parasite's ability to hide, could stop the progression of the disease and prevent heart-related deaths in those with persistent *T. cruzi* infections.

DOI: https://doi.org/10.7554/eLife.34039.002

In addition to classical genetic-based drug resistance, it has become increasingly appreciated that in bacterial infections, 'persister' cells develop that are transiently unaffected by chemotherapy, due in most cases to metabolic dormancy (reviewed in [*Harms et al., 2016*]). Such persisters can arise spontaneously in the presence or absence of stress signals and can reemerge after the termination of environmental challenges – such as antibiotic treatment.

In this study, we document the occurrence of dormancy in the protozoan *T. cruzi* and link this phenomenon to the resistance of this parasite to otherwise highly effective and cytotoxic drugs. We show that both in vitro and in vivo, *T. cruzi* exhibits a biphasic kill curve, with the rapid death of most parasites but the persistence of transiently non-replicating intracellular parasites that are resistant to high level and prolonged drug exposure.

## Results

The anti-*T. cruzi* drugs BZN and NFX are normally dosed in animals (including humans) for 30–60 days, although shorter treatment times can occasionally be curative (*Bustamante et al., 2014*; *Viotti et al., 2011*). This general requirement for a long treatment course and the variability in outcome is inconsistent with the documented rapid trypanocidal activity of BNZ for *T. cruzi* in vitro (*Moraes et al., 2014*), although to our knowledge, kill rates had not previously been determined in vivo. To directly assess the activity of BNZ in vivo, we first established a localized infection of *T. cruzi* in the footpads of mice and then used a transgenic luciferase reporter to monitor the loss of parasites subsequent to a single oral dose of BNZ. We first determined that *T. cruzi* trypomastigotes delivered in the footpad or skin, primarily infect host cells at the infection site (rather than distributing systemically) and expand as amastigotes within these host cells during the subsequent ~96 hr (Padilla, et al., in preparation). Administration of a single oral dose of BZN reduced intracellular amastigotes at the initial infection site by ~90% within 48 hr (*Figure 1A–C*). A similarly rapid reduction in parasite load occurs in mice with established systemic infections within a few days after the initiation of oral BZN treatment (*Figure 1D–F*). In this later experiment we used multiple *T. cruzi* isolates, including two (colombiana and ARC-0704) previously determined to be relatively more

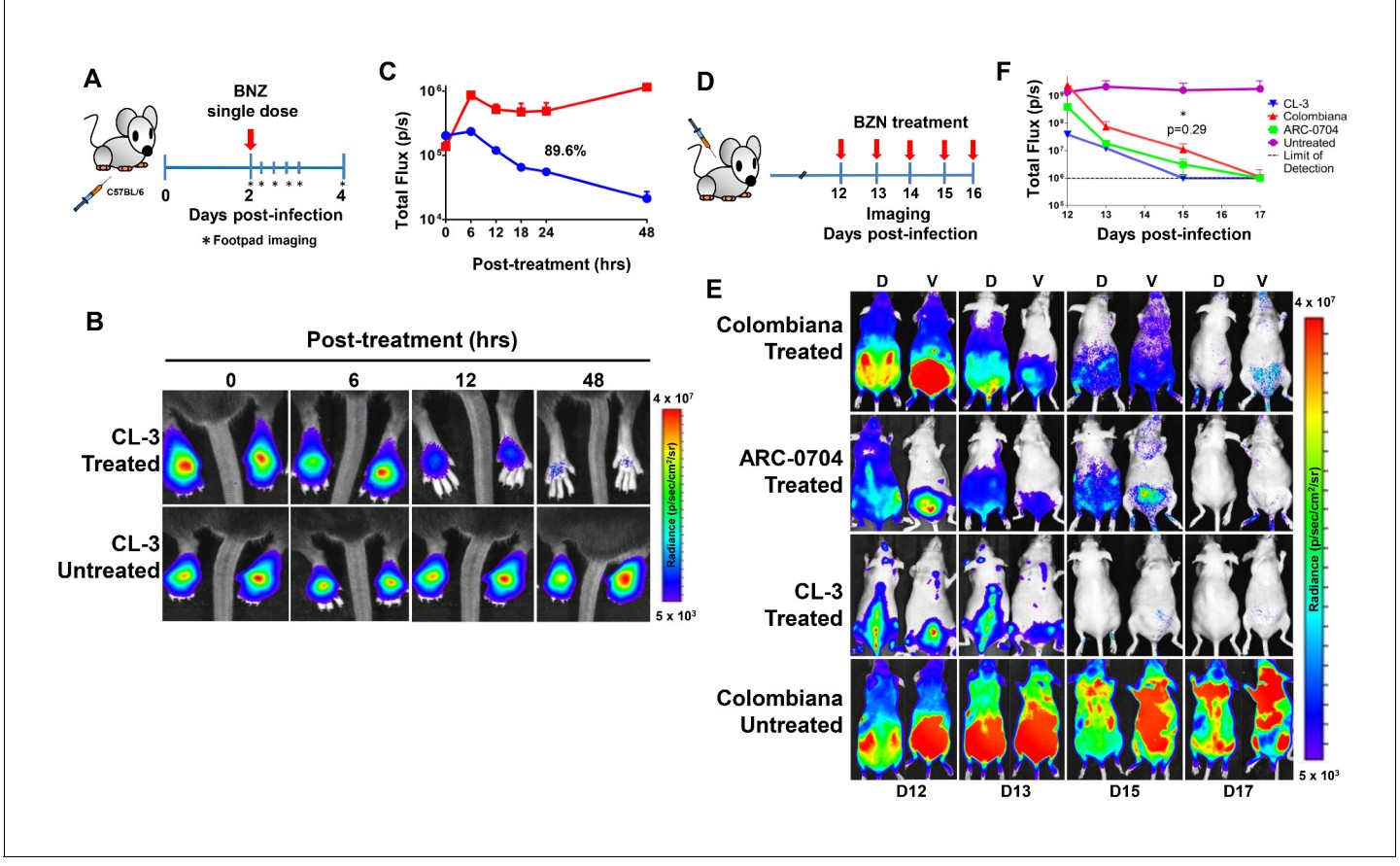

**Figure 1.** Rapid decrease of parasite load following short-term BZN treatment. (A) Schematic experimental protocol. C57BL/6 mice were infected in the hind footpads with $2 \times 10^5$ luciferase-expressing *T. cruzi* trypomastigotes of the CL-3 strain. A single oral dose of BZN (100 mg/kg) was administered 2 days post-infection. Cohorts of mice were maintained as untreated controls. Parasite bioluminescence following D-luciferin injection was measured at 6, 12, 18, 24 and 48 hr after BZN dosing. (B) Representative images showing footpad bioluminescent signal 2 days post-infection and at 6, 12 and 48 hr post-treatment. The heat map is on a $\log_{10}$ scale and indicates the intensity of bioluminescence from low (blue) to high (red). (C) Quantification of footpad signal. Each data point represents the mean of 12 footpads bioluminescence from six mice expressed on a logarithmic scale. After subtraction of the background signal, total flux measurements of photons per second (p/s) were quantified. A statistically significant difference (p=0.004) was found between treated (blue) and untreated (red) groups. (D) Protocol to measure systemic parasite load by bioluminescence before and during daily dosing with BZN. Cohorts of six SKH-1/B6 (hairless C57BL/6) mice were infected i.p. with $5 \times 10^5$ colombiana, ARC-0704 or CL-3 luciferase-expressing *T. cruzi* isolates. Oral BZN-treatment (100 mg/kg/day) was initiated on day 12 and continued until day 17 post-infection. (E) Dorsal (D) and ventral (V) images of individual representative infected mice. (F) Quantification of whole animal ventral bioluminescence. Dashed line indicates detection threshold determined as the mean plus two standard deviation of background bioluminescence of uninfected mice. Untreated control animals infected with all the strains showed similar high parasite levels, representative images (E) and data (F) for the group infected with colombiana strain are shown. No statistical differences were observed in the rate of parasite decrease between strains (p=0.29). Results are representative of three independent experiments with six infected animals per parasite strain group.

DOI: https://doi.org/10.7554/eLife.34039.003

resistant to BZN in vivo (*Bustamante et al., 2014*). The rate of parasite clearance, approaching the limit of detection by imaging, was similar for all three strains and in the case of the colombiana infection, the 5 days of treatment resulted in a > 3 $\log_{10}$ reduction in systemic parasite load. Thus, consistent with previous in vitro assessments, BZN rapidly and highly effectively kills both 'resistant' and 'susceptible' *T. cruzi* strains in vivo.

While these experiments demonstrate that BZN treatment rapidly reduces *T. cruzi* numbers by 10- to 1000-fold over just a few days, we know from previous studies that such short-term treatments, and indeed treatments as long as 60 days, often fail to completely clear *T. cruzi* infection (*Bustamante et al., 2014*). This disconnect implicates a mechanism by which *T. cruzi* can resist highly effective trypanocidal drugs for an extended period of time. To investigate whether dormant/non-

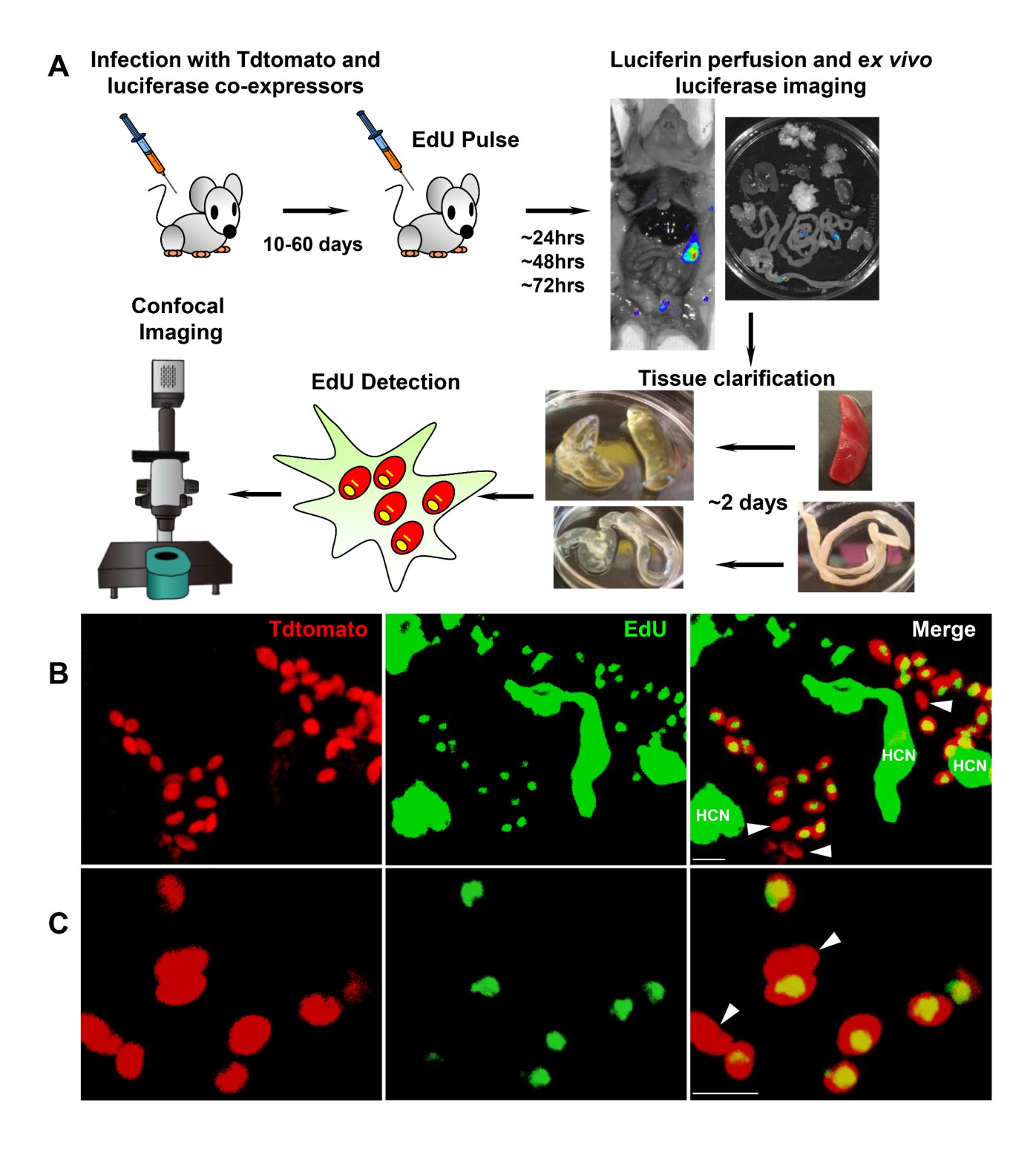

**Figure 2.** Rare amastigotes fail to incorporate EdU in chronically infected mice. (**A**) Experimental protocol for assessing proliferation of amastigotes in established in vivo infections. C57BL/6 mice were infected with $2.5 \times 10^5$ trypomastigotes of colombiana *T. cruzi* strain co-expressing fluorescent (Tdtomato) and luminescent (luciferase) reporter proteins. Sixty days post-infection, mice were injected i.p. with EdU and sacrificed 24, 48 or 72 hr after injection. Mice were perfused with PBS and luciferin and ex vivo bioluminescence imaging of selected tissues were performed to identify parasite foci. *Figure 2 continued on next page*

*Figure 2 continued*

Luciferase-positive thick tissue sections were clarified and Tdtomato + parasites and EdU incorporation was detected by confocal microscopy. Colocalization of Tdtomato (red) and EdU (green) positive signals identifies proliferating amastigotes (yellow nuclei) from (B) skeletal and (C) adipose tissue. Arrows indicate rare red only (non-proliferating) amastigotes. Scale bars, 5 µm. HCN = EdU positive host cell nuclei. Results are representative of three independent experiments using groups of 2–3 mice.

DOI: https://doi.org/10.7554/eLife.34039.004

The following figure supplements are available for figure 2:

**Figure supplement 1.** Detection of proliferating amastigotes in acutely infected mice.

DOI: https://doi.org/10.7554/eLife.34039.005

**Figure supplement 2.** Detection of non-proliferating amastigotes in established infections.

DOI: https://doi.org/10.7554/eLife.34039.006

replicating forms of *T. cruzi* might be present in infected hosts and possibly contributing to this requirement for long treatment regimens, we used in situ labeling with nucleotide analog 5-ethynyl-2'-deoxyuridine (EdU) to determine the replication status of parasites within tissues of infected mice. We first established the in vivo EdU labeling protocol in acutely infected mice with high tissue parasite load, finding that the vast majority of amastigotes incorporated EdU during a 24 hr period after EdU injection (*Figure 2—figure supplement 1*). We then applied this protocol to mice with established chronic *T. cruzi* infections and again found that most intracellular amastigotes incorporated EdU over the 24–72 hr of exposure. However a small fraction of intracellular amastigotes failed to show evidence of replication over the 72 hr after EdU injection (*Figure 2—figure supplement 2*). Interestingly, these apparently non-replicating amastigotes were within host cells in which the other amastigotes were clearly replicating, suggesting that there is a heterogeneity of amastigotes in infected cells. Since the low parasite load in chronically infected mice makes it highly unlikely that a single host cell is infected by more than one trypomastigote, this heterogeneity in replication appears to establish within host cells from a single infecting parasite *Figure 2*.

To examine in greater detail the replication pattern of *T. cruzi* in host cells, we moved to in vitro systems wherein large numbers of infected cells can be readily monitored longitudinally during infection and the continuous exposure to EdU can be assured. As in the in vivo infection, a small fraction of amastigotes in host cells in vitro consistently failed to take up EdU, even when exposed for up to 72 hr of the ~96 hr intracellular replication phase (*Figure 3*). One possible reason for the failure to replicate is that some amastigotes might die within host cells (perhaps as a result of lethal damage to DNA, defective cytokinesis, or unequal dispersal of chromosomes/organelles during prior replication, among other possible reasons). However monitoring for apoptosis of amastigotes in host cells revealed this process to be extremely rare (*Figure 3—figure supplement 1*).

Incorporation of EdU into DNA can impact DNA replication and thus cell division (*Zhao et al., 2013*). To rule out a role for EdU itself in inhibition of parasite replication, we used the cell division tracker dyes carboxyfluorescein succinimidyl ester (CFSE) and CellTrace Violet as a second method to monitor parasite replication in host cells. Both dyes couple to cellular components and are equally distributed to daughter cells during cytokinesis and with vigorous replication, the dyes are eventually diluted beyond the point of detection. *T. cruzi* trypomastigotes uniformly stain with either dye and upon infection of host cells, replicating amastigotes dilute the dye, while a fraction of amastigotes remain dye positive, indicating again the limited replication of a subset of amastigotes (*Figure 4A and B*). Amastigotes remaining dye positive after 72–96 hr were also EdU-negative (*Figure 4C*).

By continuing to monitor infected host cells over time, we observed that amastigotes that had failed to incorporate EdU during the final ~72 hr in host cells were fully competent to convert to trypomastigotes and to exit host cells, along with the previously replicating (EdU+) trypomastigotes (*Figure 4D,E*). The use of the CellTrace Violet to mark parasites that had undergone a limited number or no divisions during the infection cycle in host cells allowed us to further characterize and ultimately isolate this minority population of trypomastigotes from infected cell cultures (*Figure 4F and G*), and to assess their ability to establish a second round of infection/replication in new host cells. Not only could trypomastigotes from minimally replicating amastigotes infect new host cells, re-convert to amastigotes (*Figure 5A*), and begin a new round of replication, but progeny of this replication can also stop dividing soon after invasion and retain dye for >6 days while other progeny of this dormant amastigote divided extensively (*Figure 5B*). Thus, a small proportion of *T. cruzi* amastigotes

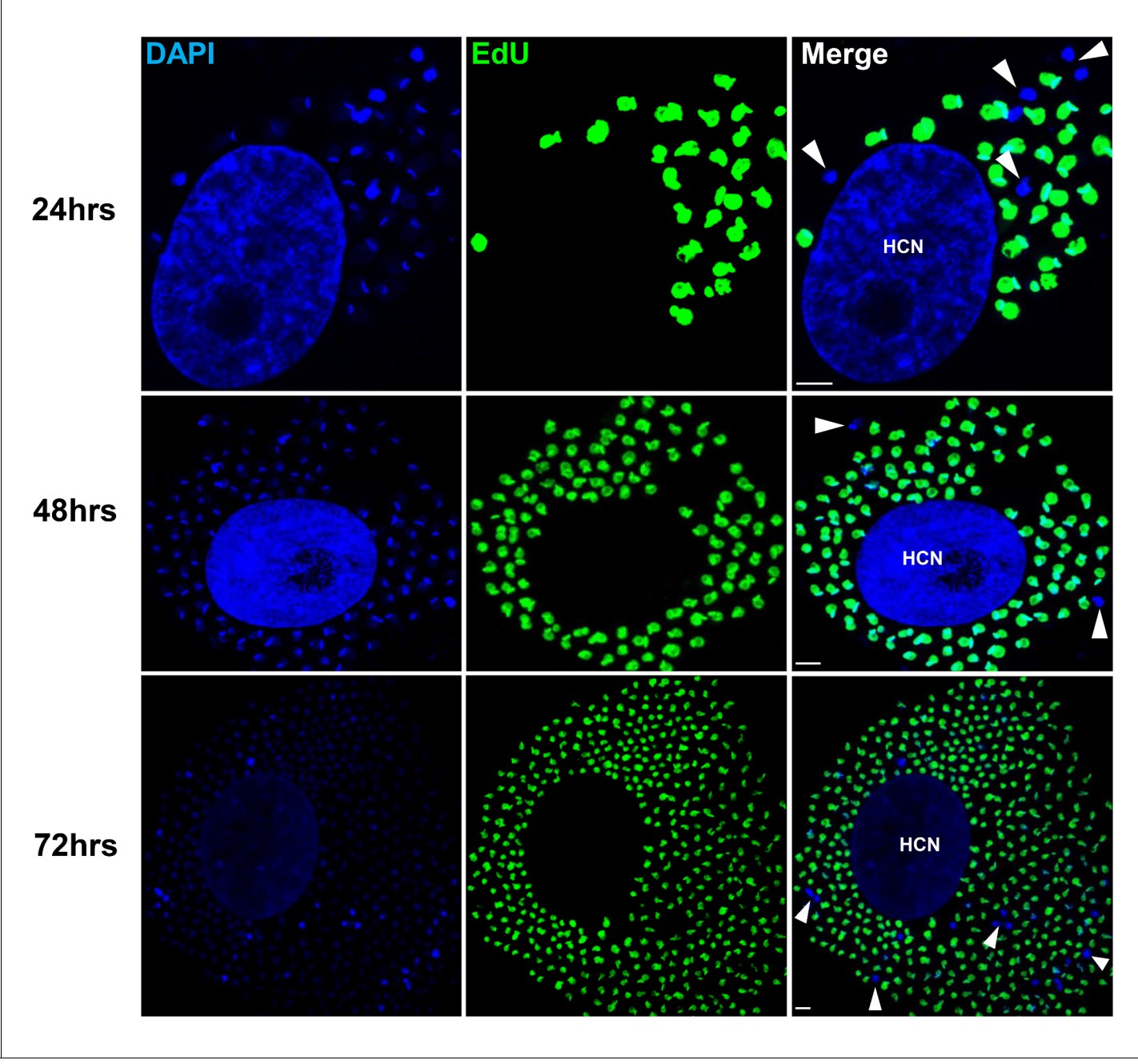

**Figure 3.** Non-replicating amastigotes are also evident in vitro. Nearly confluent monolayers of Vero cells were infected with trypomastigotes of colombiana, CL, Brazil or ARC-0704 *T. cruzi* strain. Twelve hours after infection, cultures were washed and incubated with EdU (100 μM) diluted in fresh RPMI medium. EdU was detected after an additional 24, 48 or 72 hr culture period. DAPI staining (blue) allows the identification of EdU-negative amastigotes (arrows) from the EdU-positive (green). HCN, host cell nuclei. Scale bars, 5 μm. Results are representative of three independent experiments.

DOI: https://doi.org/10.7554/eLife.34039.007

The following figure supplement is available for figure 3:

**Figure supplement 1.** Rare apoptosis of amastigotes in host cells.

DOI: https://doi.org/10.7554/eLife.34039.008

halt replication within 24 hr after host cell entry but can convert to trypomastigotes that are infection-competent and able to repeat the processes of replication and arrested replication in new host cells (*Figure 5C*).

The detection of a small subpopulation of *T. cruzi* amastigotes undergoing minimal replication through 2 rounds of host cell invasion suggests that the cessation of replication can at a minimum happen early after host cell invasion, but did not preclude a more dynamic pattern of transitioning between replication and dormancy during the infection cycle in host cells. In order to better understand the flexibility and heterogeneity of replication within host cells, we monitored the early replication of *T. cruzi* in host cells by time-lapse video, using violet dye dilution to help mark those parasites undergoing minimal cell division within the same cell where other parasites are actively dividing (Sup *Videos 1* and *2*).These videos also indicate other characteristics of non-dividing amastigotes that were apparent in other still images, for example that non-dividing amastigotes are

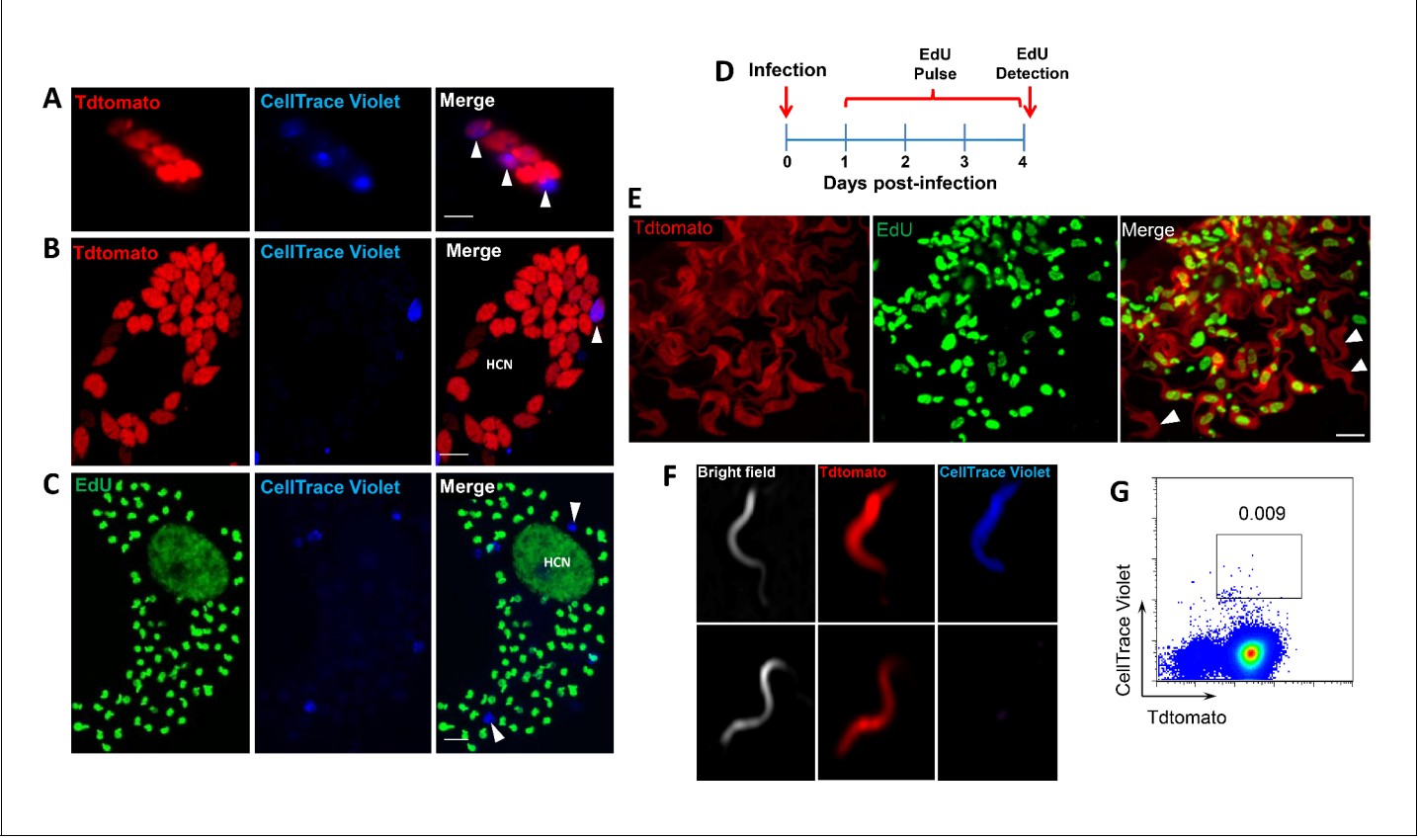

**Figure 4.** Non-proliferating amastigotes can transition to trypomastigotes. (A, B) Td-tomato expressing *T. cruzi* trypomastigotes were labeled with CellTrace Violet and used to infect Vero cells. Amastigote replication was monitored for 72–96 hr post-infection by live imaging (A) and confocal microscopy (B). Note the violet signal dilution in proliferating amastigotes and the retention of the dye in the non-replicating amastigote (arrows). (C) Vero cell cultures infected with CellTrace Violet-labeled *T. cruzi* trypomastigotes were incubated with EdU (100 μM) for 72 hr before fixation and detection of EdU incorporation. CellTrace Violet-positive amastigotes were also EdU-negative (arrow). Experiment was performed three times. (D) Schematic of the experimental protocol. *T. cruzi* colombiana strain trypomastigotes expressing Tdtomato were used to infect nearly confluent monolayers of Vero cells. Twelve hours after infection, the dishes were washed and incubated with EdU (100 μM) for 72 hr. Cultures were fixed, permeabilized and EdU (green) detection was performed just prior to trypomastigote release from host cells (~96 hr post-infection). (E) Arrowheads indicate EdU-negative trypomastigotes (red only) within host cells filled with EdU-positive trypomastigotes (green). (F, G) Violet-labeled Td-tomato-expressing trypomastigotes (colombiana or CL strains) were used to infect Vero cell cultures. Trypomastigotes released 92 hr post-infection were harvested by centrifugation and flow-imaged by ImageStream and sorted via fluorescence-activated flow sorting. Only a small subpopulation of trypomastigotes retained substantial CellTrace Violet after a single round of host cell infection. Similar results were obtained with CFSE staining of trypomastigotes (not shown). Scale bars, 5 μm. Experiment was performed a minimum of three times.

DOI: https://doi.org/10.7554/eLife.34039.009

similar in size to recently divided amastigotes and are generally Tdtomato-dim, expressing low levels of the Tdtomato fluorescent protein relative to rapidly dividing amastigotes.

These results show conclusively that within host cells infected with *T. cruzi*, a subset of amastigotes cease replication while others continue rapid division, eventually filling the host cell with amastigotes. The obvious question remaining is 'Does this cessation of replication play a critical role in the failure of drugs to effectively clear *T. cruzi* infection?' More specifically, are these replication-arrested parasites resistant to drug treatment? To directly address these questions, we infected mice with Tdtomato-expressing and CellTrace Violet-stained trypomastigotes and confirmed 2 days post-infection the presence in adipose tissue of parasites displaying both of these markers (*Figure 6A and B*). As expected based on previous experiments, over the span of the 2 day

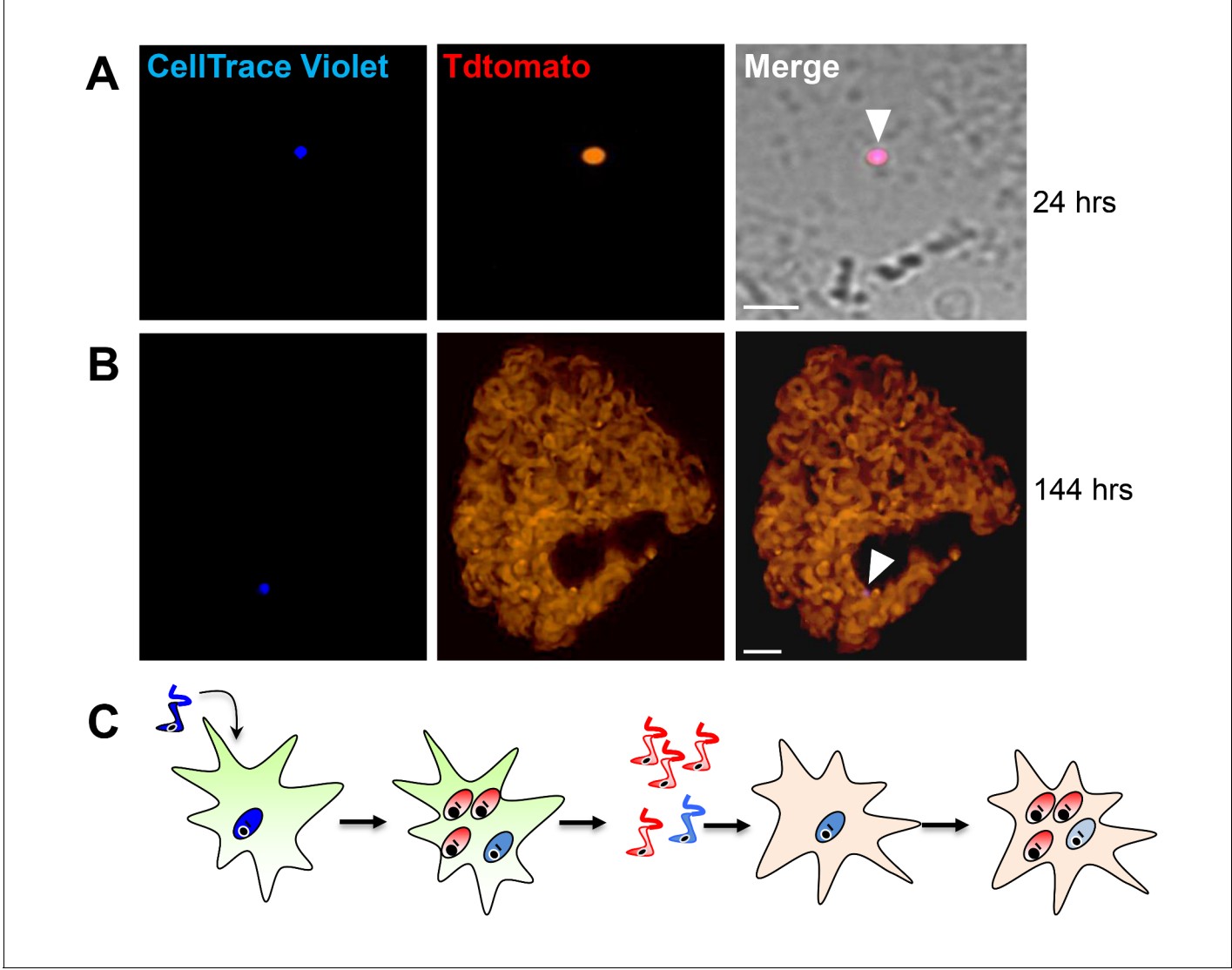

**Figure 5.** Trypomastigotes originating from non-replicative amastigotes are able to infect host cells and produce both replicating and dormant progeny. (A) Violet-positive trypomastigotes obtained as shown in *Figure 4G* were used to infect a fresh culture of Vero cells. Infected cells with CellTrace Violet-positive amastigotes were observed 24 hr post-infection (arrow). (B) Six days post-infection, a non-dividing CellTrace Violet-positive progeny was identified (arrow) in cells filled with dye-negative trypomastigotes. (C) Experiment summary: CellTrace Violet-positive trypomastigotes can produce both dormant (violet+) and actively replicating (violet-) progeny, both of which can convert to trypomastigotes. The trypomastigotes from previously dormant amastigotes can infect new host cells and repeat the process of generating both dormant and actively replicating progeny. Scale bars, 5 um. Experiments repeated twice.

DOI: https://doi.org/10.7554/eLife.34039.010

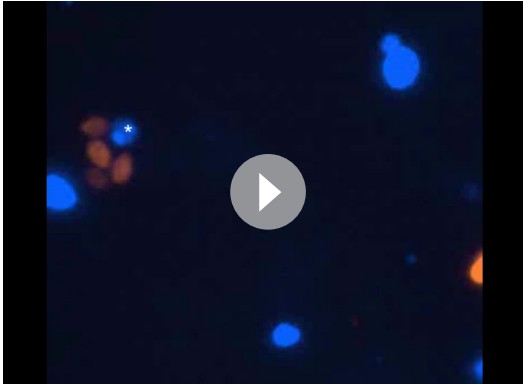
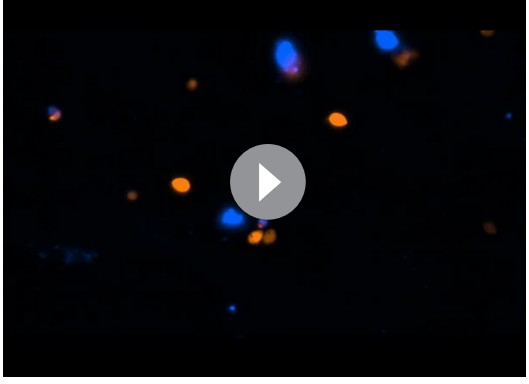

**Video 1.** Time lapse video of early stage intracellular replication of *T. cruzi* amastigotes. Cultures of HFF cells in 96 well glass bottom plates were infected with Tdtomato-expressing Cell Trace Violet labeled trypomastigotes in a ratio of 10:1 (parasites: host cells). After a5hrincubation, cultures were washed to remove extracellular parasites and imaged every 15 min for 48hr. Time lapse videos were generated spanning from 15:45 to 31:30hr. Note the dye-retaining amastigote (blue) remaining undivided while other amastigotes in the same cells (bright orange amastigotes) actively proliferate.

DOI: https://doi.org/10.7554/eLife.34039.011

**Video 2.** Time lapse video of early stage intracellular replication of *T. cruziamastigotes*. Cultures of Vero cells in 96 well glass bottom plates were infected with Tdtomato-expressing CellTrace Violet labeled trypomastigotes in a ratio of 5:1 (parasites: host cells). After a5hrincubation, cultures were washed to remove extracellular parasites and imaged every 15 min for 48hr. Time lapse videos were generated spanning from 9:30 to 47:15hr. Note the dye-retaining amastigote (blue amastigote denoted by white asterisk at the beginning of *Video 1*) remaining undivided while other amastigotes in the same cell (bright orange amastigotes) actively proliferate.

DOI: https://doi.org/10.7554/eLife.34039.012

infection, several strongly violet-positive parasites are obvious in cells along with a larger number of parasites in which the violet dye was faint or not detectable, indicating both slow or non-replicating and actively replicating progeny of the infecting trypomastigotes. On days 2 and 3, similarly infected mice were treated with BZN or left untreated and on day 4, examined for the presence of parasites. Remarkably, in the BZN-treated mice, the only parasites detected were brightly violet and present at 1 and occasionally two per infected cell (*Figure 6B* and *Figure 6—figure supplement 1*). Actively replicating parasites (violet-negative) were not detected in the BZN-treated mice while a much larger number of violet-negative and less abundant violet-positive parasites were observed in the non-treated mice. A resumption of parasite replication was evident in treated mice 9 days after the last of the 2 BNZ treatment doses, demonstrating that dormant, and therefore drug resistant amastigotes, could also resume division in vivo. To further address the drug resistance and recovery potential of dormant amastigotes in vivo, we repeated these experiments in interferon-gamma deficient mice, which lack the ability to develop immune control of *T. cruzi* infection. Non-replicating amastigotes developing in these mice survive a minimum of 20 days of BZN treatment (*Figure 6—figure supplement 2A*) and again resume replication upon cessation of BZN treatment (*Figure 6C*) and eventually spread extensively in these animals (*Figure 6—figure supplement 2B,C*).

Tracking very low numbers of persisting parasites in vivo in drug-treated, infected animals is quite difficult, so we returned to in vitro systems to examine the resistance of *T. cruzi* to drug treatment over longer time periods and the association between drug resistance and dormancy. Here we infected host cells 24 hr previously with Tdtomato-expressing and CellTrace Violet-labeled parasites and then added BZN at 10X the $IC_{50}$ for up to 30 days of culture before washing out the drug. Parasite recovery following drug treatment/washout was monitored globally by whole-well fluorescence detection of the Tdtomato reporter, and at the individual cell level by microscopy (*Figure 7*). In the absence of treatment, parasite numbers peak near the time of the completion of the first round of intracellular replication (~day four after infection). In all cases, even following 30 days of high dose BZN treatment, parasites can rebound and begin replication after BZN is removed. This rebound can often be observed in the whole well Tdtomato fluorescence readings (*Figure 7A*), and in all

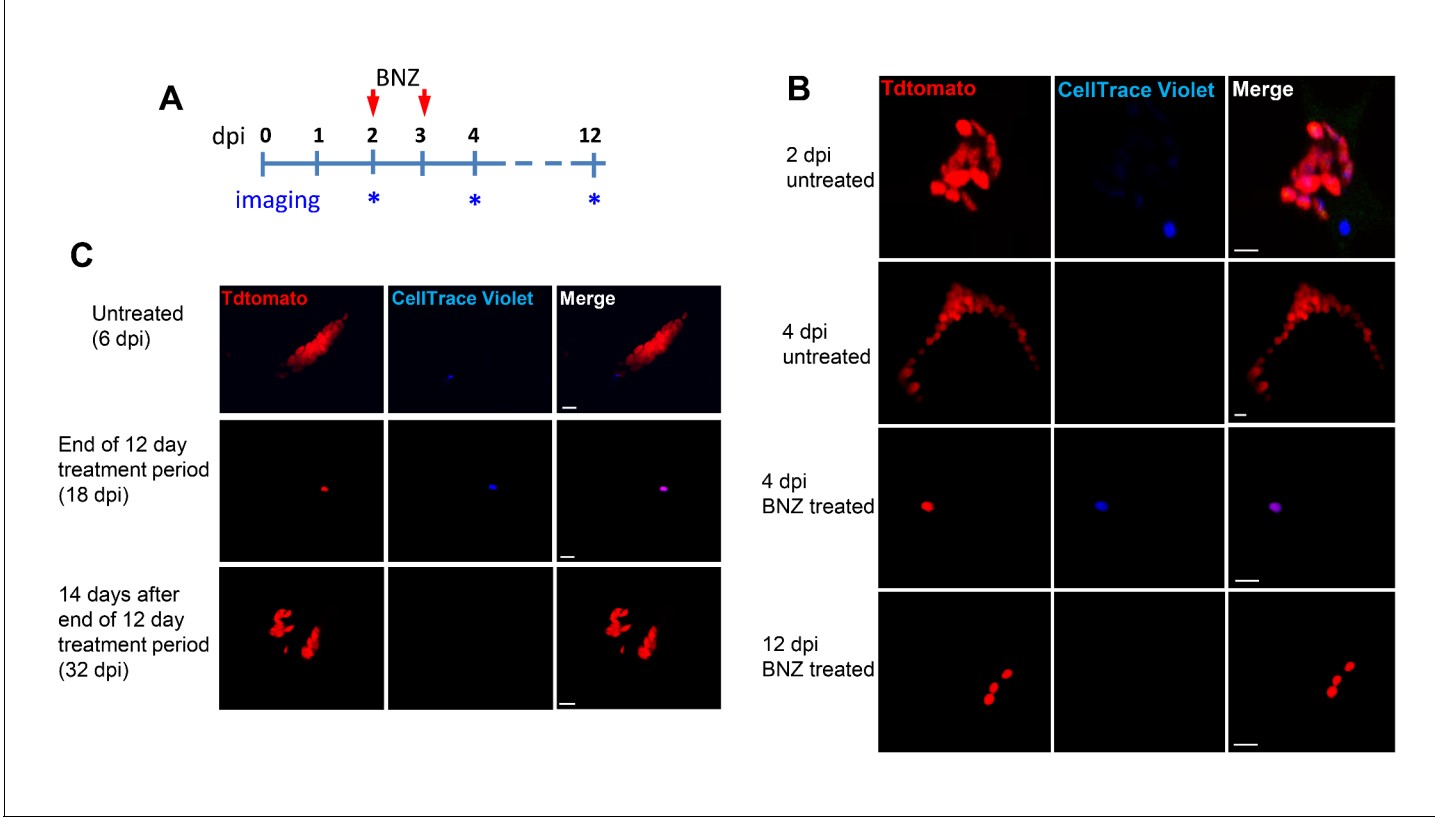

**Figure 6.** Only dormant amastigotes are resistant to short-term in vivo BZN treatment. (**A**) Schematic of experimental protocol. (**B**) C57BL/6 mice were infected i.p. with 5 × 10⁶ Td-Tomato-expressing and CellTrace Violet-labeled trypomastigotes and orally treated with BZN (100 mg/kg/day) on day 2 and 3 post-infection or left untreated. At the indicated time points, mice were euthanized and adipose tissue was excised, fixed and processed for confocal imaging. Images are representative of three independent experiments with 3–4 animals per group. (**C**) Infection with 1.5 × 10⁷ Td-Tomato-expressing and CellTrace Violet-labeled trypomastigotes was established for 6 days in IFN-g deficient mice before initiation of daily treatment with BZN (100 mg/kg) for 12 days. Peritoneal adipose tissue was harvested from mice on day 6 (prior to treatment), at the end of the 12 day treatment period on day 18 post-infection, or 14 days after the end of treatment (32 dpi). The peritoneal adipose tissue sample from each mouse is ~0.3 g of tissue spread over a surface of ~1.6 cm² when mounted for microscopic analysis. In tissue harvested on day 6 of infection (prior to treatment), infected cells can be observed within 15 min of scanning. Following 12 days of treatment this same amount of tissue must be exhaustively scanned for up to 3 hr per sample to detect between 4 and 5 infected cells per sample. Images are representative of two independent experiments with four animals each.
DOI: https://doi.org/10.7554/eLife.34039.013

The following figure supplements are available for figure 6:

**Figure supplement 1.** Residual CellTrace Violet-labeled amastigotes after BZN treatment.
DOI: https://doi.org/10.7554/eLife.34039.014

**Figure supplement 2.** Dormant amastigotes resist up to 20 days of BZN treatment in vivo and are able to rebound after treatment cessation.
DOI: https://doi.org/10.7554/eLife.34039.015

cases, low numbers of parasites with a range of violet dye retention are evident in cultures during drug treatment (*Figure 7B*), as well as resumption of low level to vigorous replication after drug washout (*Figure 7C*). The resistance of *T. cruzi* to drug treatment due to amastigote dormancy was not restricted only to BNZ, as members of a class of newly developed oxaborale compounds currently under preclinical development also failed to overcome dormancy-dependent resistance in vitro (*Figure 7—figure supplement 1*). Further, *T. cruzi* lines rebounding after 30 days of in vitro exposure to BZN had not developed stable resistance to the compound as the IC₅₀ of BZN on this population remained unchanged from that of the pre-exposed population (*Figure 7—figure supplement 2*).

Thus, *T. cruzi* amastigotes regularly and spontaneously cease replication and in that dormant state are resistant to otherwise highly effective trypanocidal drugs for extended periods of time.

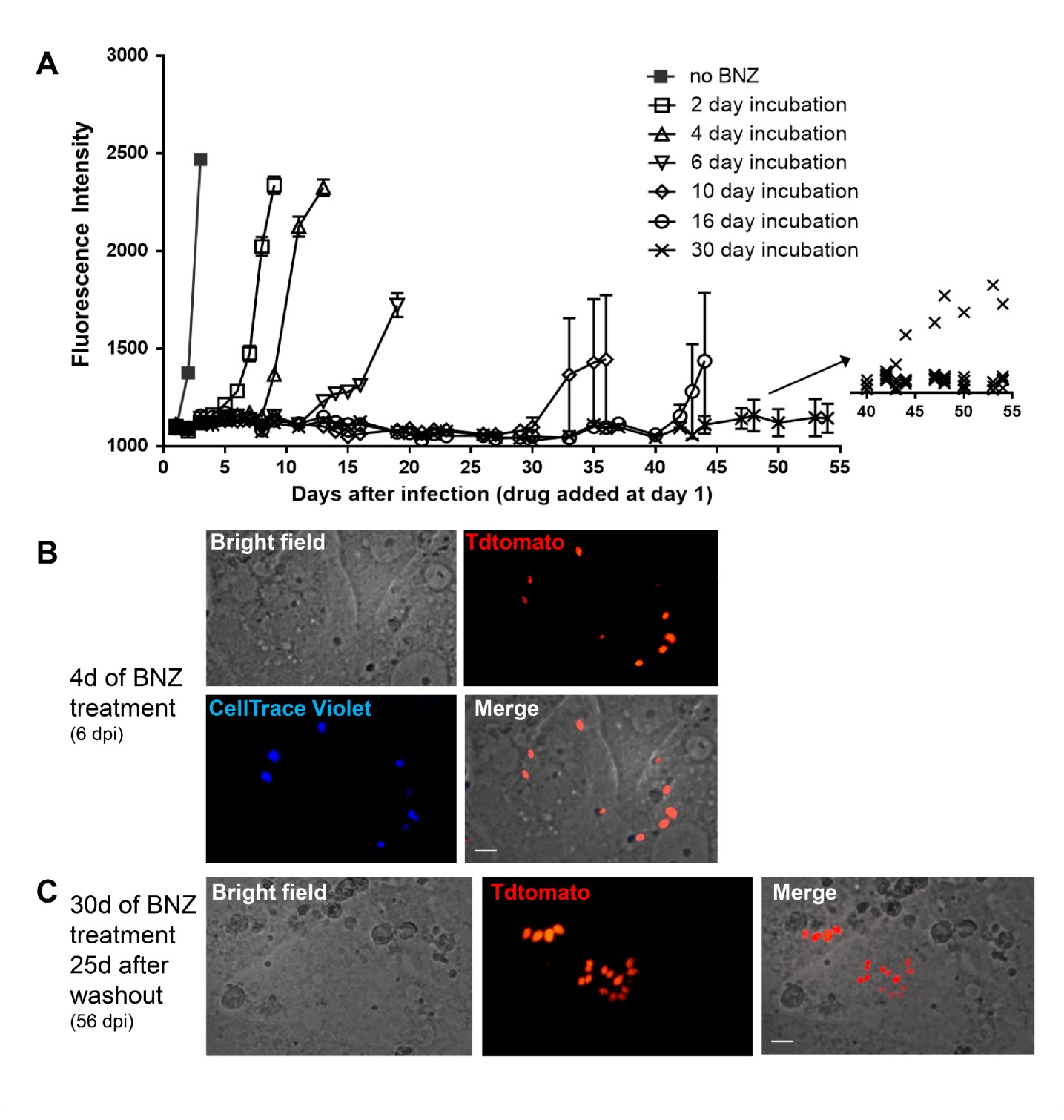

**Figure 7.** Dormant amastigotes are resistant to extended BNZ treatment in vitro. Vero cell cultures were infected with Tdtomato-expressing trypomastigotes of colombiana or CL strains (5:1 ratio parasites:cells) 24 hr prior to addition of BNZ (10 uM). At various times post-treatment, BNZ was removed from the cultures and the subsequent rebound of parasites surviving the drug treatment was determined by whole well fluorescence intensity reading (A) and in vitro live imaging (B, C). The retention of the CellTrace Violet label during the course of BNZ treatment indicates that the BNZ-resistant parasites are dormant (B) and capable of replication after drug washout (C). Results are representative of three independent experiments with six replicates per condition. Arrow to inset in (A) shows reading of individual replicate wells.

DOI: https://doi.org/10.7554/eLife.34039.016

*Figure 7 continued on next page*

*Figure 7 continued*

The following figure supplements are available for figure 7:

**Figure supplement 1.** Dormant amastigotes are resistant to extended oxaborale treatment in vitro.

DOI: https://doi.org/10.7554/eLife.34039.017

**Figure supplement 2.** Amastigotes surviving 30 days in vitro BZN treatment have unaltered susceptibility to BZN.

DOI: https://doi.org/10.7554/eLife.34039.018

These dormant parasites are capable of re-initiating replication after >30 days of drug exposure and perhaps much longer after the initiation of dormancy.

## Discussion

Although *T. cruzi* infection is one of the highest impact infectious disease in the Americas, including in the United States, effective prevention, control and treatment methods are virtually non-existent. A rare bright spot in potential interventions for Chagas disease is the increased availability of drugs such as BZN, which have substantial curative efficacy. However, although BZN and NFX have been in use for decades, have relatively clear mechanisms of action and do not appear to readily encourage development of genetic-based drug resistance, their unpredictable treatment outcome coupled with their modest to severe side effects has limited their use primarily to acute, childhood, or immuno-suppression-exacerbated infections. This situation leaves largely untreated the millions of individuals with chronic *T. cruzi* infection, many of whom will eventually develop chagasic cardiomyopathies. Furthermore, human clinical trials of new therapies judged to be promising based upon limited studies in experimental models have been colossal failures (*Molina et al., 2014*; *STOP-CHAGAS Investigators et al., 2017*) and ClinicalTrials.gov identifiers NCT01489228, NCT02498782, and NCT01162967), demonstrating these compounds to be clearly inferior to BNZ and NFX.

This study initially sought to determine why compounds like BZN that can, and often do, provide sterile cure in *T. cruzi* infection, also frequently fail. This and previous studies have largely ruled out several mechanisms as primary reasons for the variable efficacy of BZN. Although there is parasite strain variation with respect to BZN susceptibility in vivo, this variation does not appear to be due to direct resistance to the trypanocidal activity of BZN (*Figure 1*). Parasite lines obtained from failed drug treatments are no more resistant to BZN than the pre-treatment parasites (Sup *Figure 7*), and although BZN resistance can be induced by drug selection in vitro, such partially resistant lines are crippled with respect to initiating and maintaining an infection (*Mejia et al., 2012*). Thus resistance to BZN treatment is relative, not complete, is not easily selectable, and is not associated with any specific genetic lineage of *T. cruzi* nor with host genetics.

The results of the current study firmly connect drug treatment failure in *T. cruzi* infection to the presence of a previously unrecognized player in this parasite's life cycle, the transiently dormant amastigote. We denote these stages as dormant based primarily on their failure to replicate over an extended period of time and their resistance to trypanocidal compounds over >30 days of exposure. These dormant amastigotes appear at a consistently detectable frequency and are present in many infected host cells in vitro and in vivo. Dormancy occurs spontaneously and is often observed soon after host cell infection – as shown by the ability of parasites to retain cell division monitoring dyes through several rounds of host cell invasion and expansion. However dormancy may also occur at other points throughout the 4–5 day replication cycle within host cells, as suggested by the presence of multiple EdU-negative amastigotes in heavily infected cells and supported by time lapse imaging of infected cells. Importantly, these dormant parasites are not permanently arrested, as they resume replication days to weeks after entering dormancy (*Figure 7*). These dormant amastigotes also readily respond to the cues that drive conversion to trypomastigote forms within host cells, a process that appears to occur only when the host cell is nearly bursting with amastigotes. Candidates for the inducer of stage conversion would include depleted host cytoplasmic components or byproducts of parasite metabolism that could accumulate inside the host cell. Thus the dormant amastigotes have two potential fates besides continued dormancy: to re-initiate replication like conventional amastigotes – likely to occur when there are none or a limited number of other amastigotes in the cell - or to convert to trypomastigotes in concert with the actively replicating amastigotes undergoing that

process within the same cells. An extremely rare outcome for amastigotes in host cells is death, as determined by the paucity of degrading amastigotes detected in host cells (Sup *Figure 3*).

Dormancy is well-studied in bacteria and plays a critical role in their adaptation to changing environments. This phenomenon had been missed by previous investigations in *T. cruzi* likely because extended dormancy (which would be required to survive during weeks of drug treatment) is relatively rare. In vitro studies often report very high efficiency of drugs such as BZN to kill >99% of *T. cruzi* amastigotes – and the assumption has been that this was in fact 100% killing (*Moraes et al., 2014*), but this is clearly not the case. To our knowledge, this is the first report of extended dormancy in an otherwise replicating stage of any trypanosomatid. In the related protozoan *Leishmania* sp, intracellular amastigotes exhibit greatly reduced proliferation and metabolism relative to the extracellular promastigote forms (*Jara et al., 2017*) and have been described as 'semi-quiescent' but steadily replicating in vivo with a doubling time of 12 days for *L. mexicana* (*Kloehn et al., 2015*). Studies in *L. major* by Mandell and Beverley documented slow growing (~60 hr doubling time) and even slower dividing, BrdU-non-incorporating intracellular amastigotes in mouse infections but the technical limitations of the labeling procedure prevented a definitive conclusion of dormancy (*Mandell and Beverley, 2017*). The dormancy we describe here for *T. cruzi* is clearly very distinct from but perhaps most similar to the scantly studied arrested hypnozoite liver stages of *Plasmodium vivax*. Hypnozoites appear responsible for relapse of vivax malaria even years after the initial infection (*Markus, 2012*), although studies in human liver chimeric mice suggest that hypnozoites may actually not be fully metabolically inactive (*Mikolajczak et al., 2015*).

Cyst stages of apicomplexans such as *Toxoplasma gondii* and *Cryptosporidium sp* exhibit greatly reduced if not fully quiescent metabolism, consistent with a life cycle stage that is resistant to potentially harsh environmental conditions associated with transmission to a new host. For *T. cruzi* it is much less clear what the evolutionary advantage of dormancy is since transmission depends on putting extracellular parasites into the mammalian host bloodstream for ingestion by its blood feeding reduviid vector. Although we show that dormancy provides a degree of protection from drug-mediated clearance, drug resistance itself cannot be the selective force for this developmental pathway as all the *T. cruzi* lines we have examined, including ones that had not been previously exposed to these or other drugs, generate dormant amastigotes.

If dormancy is programmed and essential for *T. cruzi* survival, then the property would seem most likely associated with evasion of host immune responses, the major stressor with which *T. cruzi* amastigotes have to contend. *T. cruzi* induces a robust and highly effective immune response that greatly reduces parasite transmission to insect vectors. Although it is relatively rare, this immune response can lead to complete resolution of the infection (*Dias et al., 2008*; *Tarleton, 2013*; *Viotti et al., 2006*). Dormancy could facilitate parasite persistence if some trypomastigotes entering host cells immediately went quiescent and as a result were not detected by the immune system. Such persisters could then 'wake-up' and begin replication weeks or months later. But those reawakened parasites would likely remain low in numbers as they would have to contend with the now established anti-*T. cruzi* immune response in order to expand and have an increased chance of being transmitted. Thus the selective pressure for development or retention of a dormancy pathway for this purpose alone seems very low.

Alternatively, dormancy might be a consequence of the biology of *T. cruzi*, rather than an evolutionarily selected process for low-level persistence. *T. cruzi* very rarely undergoes sexual recombination but instead depends heavily on gene amplification and recombination, as well as horizontal gene transfer, for generating genetic diversity. The resorting of genetic material likely takes place in amastigotes, one of the two replicating stages in the *T. cruzi* life cycle. Parasites that enter dormancy soon after host cell invasion could be pre-recombination, 'stem-like' cells that would be useful should the genetic recombination events in the replicating parasites go awry. Alternatively, parasites might enter dormancy in order to sort out the result of recombination events before proceeding with additional rounds of replication.

Whether stochastic or programmed, the mechanisms for entering and exiting the dormant state by *T. cruzi* is a complete black box. Hints about the process may come from understanding the events involved in transitioning from replicating stages to non-replicating stages in *T. cruzi* (e.g. epimastigotes to infective metacyclic trypomastigote and amastigote to blood stage trypomastigote) or in the related African trypanosome *Trypanosoma brucei*, which although lacking an intracellular stage, makes several well-characterized transitions from replicating to non-replicating forms.

The biggest practical message from the current study is that dormancy in the mammalian infection cycle of *T. cruzi* is key to the failure of current drug treatments for *T. cruzi* infection. While other factors, including the differential tissue tropism of parasite strains and tissue distribution of potential drugs, certainly also impact treatment outcomes, only dormancy has been definitively linked. Given this new understanding of dormancy and its impact on drug treatment, simply identifying more compounds that effectively kill metabolically active parasites is not likely on its own to be the solution to achieving more dependable cure in *T. cruzi* infection. The development of new assays that can screen for compounds capable of overcoming dormancy should also be part of the paradigm for identifying new, more effective compounds. The use of current drugs in extended, less intensive dosing routines (*Álvarez et al., 2016*; *Bustamante et al., 2014*) that extend beyond the dormancy potential of *T. cruzi*, in place of the current highly intensive regimens, should also be further explored.

## Materials and methods

**Key resources table**

| Reagent type (species) or resource | Designation | Source or reference | Identifiers | Additional information |
|---|---|---|---|---|
| Strain, strain background | Trypanosoma cruzi | | RRID:NCBITaxon:5693 | |
| Strain, strain background (*Trypanosma cruzi*, colombiana) | TdTomato and luciferase coexpresors; Luciferase- and Tdtomato-expressing parasites. | this paper | NA | Cotransfected with pTREX-Luciferase and pTREX-Td-tomato plasmids. |
| Strain, strain background (*T. cruzi*, colombiana) | Colombiana strains expresing luciferase | this paper | NA | Transfected with the pTREX-Luciferase plasmid. |
| Strain, strain background (*T. cruzi*, CL3) | CL-3 strain expresing luciferase | this paper | NA | Transfected with the pTREX-Luciferase plasmid. |
| Strain, strain background (*T. cruzi*, ARC0704) | ARC-0704 strain expresing luciferase. | this paper | NA | Transfected with the pTREX-Luciferase plasmid. |
| Strain, strain background (*T. cruzi*, colombiana) | TdTomato expresing parasites. | this paper | NA | Transfected with pTREX-Tdtomato plasmid. |
| Cell line | VERO 76 | American Type Culture Collection (ATCC) | CRL 1587; RRID:CVCL_0603 | |
| Cell line | Human Foreskin Fibroblasts (HFF) | other | N/A | HFF cells were a gift from Dr. D. Etheridge (University of Georgia) |
| Recombinant DNA reagent | pTREX-Luciferase (plasmid) | *Canavaci et al., 2010*. PMID: 20644616 | NA | Addgene 48337 |
| Recombinant DNA reagent | pTREX-Tdtomato (plasmid) | *Canavaci et al., 2010*. PMID: 20644616 | NA | Addgene 47975 |
| Commercial assay or kit | Click-iT EdU Imaging Kit | ThermoFisher Scientific, Waltham, MA | C10337 | Cell proliferation detection kit |
| Commercial assay or kit | Click-iT TUNEL Alexa Fluor 647 imaging assay | ThermoFisher Scientific, Waltham, MA | C10247 | Cell apoptosis detection kit |
| Commercial assay or kit | CellTrace Violet fluorescent dye | ThermoFisher Scientific, Waltham, MA | C34557 | Cell proliferation detection kit |
| Commercial assay or kit | CFSE fluorescent dye | ThermoFisher Scientific, Waltham, MA | C34554 | Cell proliferation detection kit |
| Chemical compound, drug | D-luciferin | PerkinElmer, Waltham, MA | 122799 | luciferase substract reagent |
| Chemical compound, drug | CUBIC clarifying solution | other | NA | 25 % N,N,N′,N′-Tetrakis (2-hydroxypropyl)ethylenediamine; 25% urea; 15% Triton X-100 and distilled water (*Susaki et al., 2015*) |

*Continued on next page*

*Continued*

| Reagent type (species) or resource | Designation | Source or reference | Identifiers | Additional information |
|---|---|---|---|---|
| Chemical compound, drug | Benznidazole (N-benzyl-2 -nitro-1-imidazolacetamida) | LAFEPE medicamentos. Brazil; Aesica Pharmaceutical, United Kingdom | BZN | |
| Chemical compound, drug | DAPI stain 4',6-diamidino-2 -phenylindole dihydrochloride | ThermoFisher Scientific, Waltham, MA | 122799 | |
| Software, algorithm | Living Image software v4.3 | Xenogen, Alameda, CA | RRID:SCR_014247 | |
| Software, algorithm | GraphPad 5.0 Prism v5.0 | GraphPad Software, La Jolla California USA, | RRID:SCR_002798 | |
| Other: Mice strains | C57BL/6NCr mice | Charles River Laboratories | C57BL/6NCrl - strain code 027 | |
| | IFN-γ knockout mice | The Jackson Laboratory | B6.129S7-Ifngtm1Ts/J - stock No 002287 | |
| | SKH-1 mice | other | NA | The SKH-1 'hairless' mice backcrossed to C57BL/6 were a gift from Dr. Lisa DeLouise (University of Rochester). |

## Mice

C57BL/6NCr and IFN-γ knockout (B6.129S7-Ifngtm1Ts/J) mice were purchased from Charles River Laboratories and The Jackson Laboratory respectively. The SKH-1 'hairless' mice backcrossed to C57BL/6 were a gift from Dr. Lisa DeLouise (University of Rochester). All mice were maintained in the University of Georgia Animal Facility under specific pathogen-free conditions. This study was carried out in strict accordance with the Public Health Service Policy on Humane Care and Use of Laboratory Animals and Association for Assessment and Accreditation of Laboratory Animal Care accreditation guidelines. The protocol was approved by the University of Georgia Institutional Animal Care and Use Committee.

## Parasites and host cells

To generate bioluminescent parasites, CL-3, ARC-0704 and Colombiana *T. cruzi* strains were transfected with the pTREX-Luciferase-Neo plasmid, generated by cloning of the firefly luciferase gene from the Luciferase-pcDNA3 plasmid (gift from William Kaelin; Addgene plasmid # 18964) into the multi-cloning site of the pTREX plasmid (*Vazquez and Levin, 1999*). Colombiana bioluminescent parasites were transfected with pTREX-Td-tomato plasmid (*Canavaci et al., 2010*) to generate parasites co-expressing bioluminescent and fluorescent markers. Transfection and selection conditions were as previously described (*Peng et al., 2014*). For single-cell cloning, parasites were deposited into a 96-well plate to a density of 1 cell/well using a MoFlow XDP (Beckman Coulter, Hialeah, FL) cell sorter and cultured in 250 µl LDNT supplemented with 300 mg/ml G418. Selected clones were screened for luciferase activity as previously described (*Lewis et al., 2014*) and a highly luciferase-expressing clone was selected for in vivo experiments. To obtain metacyclic forms, epimastigotes were submitted to stress in triatome artificial urine (TAU) medium for 2 hr. Then, parasites were incubated in complemented TAU medium (TAU3AAG))(*Bourguignon et al., 1998*) for 6–7 days, at which time transformation to metacyclic trypomastigote was maximal. Vero cells were obtained from the American Type Culture Collection (Manassas, VA) and cultured in RPMI 1640 medium with 10% fetal bovine serum. Human foreskin fibroblast (HFF) cells were a gift from Dr. Drew Etheridge (University of Georgia) and were maintained in DMEM media supplemented with 10% fetal bovine serum. Cultures were maintained in a humid atmosphere containing 5% $CO_2$ at 37°C and tested periodically for mycoplasma contamination.

## In vivo imaging

Luciferase- and Tdtomato-expressing parasites were used to assess in vivo parasite replication and clearance in mice. For bioluminescent detection, mice were injected via an intraperitoneal (i.p.) route

with D-luciferin (150 mg/kg; PerkinElmer, Waltham, MA) and anesthetized using 2.5% (vol/vol) gaseous isofluorane in oxygen prior to imaging on an IVIS 100 imager (Xenogen, Alameda, CA) as previously described (Canavaci et al., 2010). Quantification of bioluminescence and data analysis was performed using Living Image v4.3 software (Xenogen).

## In vitro infection and EdU detection

EdU incorporation into amastigote DNA was detected using the Click-iT EdU Imaging Kit (Thermo-Fisher Scientific, Waltham, MA) following manufacturer specifications. Briefly, Vero cells ($1 \times 10^5$ cells) were plated onto sterile glass-bottom 35 mm petri dishes and incubated overnight ON at 37°C, 5% CO2, in RPMI-1640 medium plus 10% fresh fetal bovine serum (FBS). Trypomastigotes of the colombiana, CL, Brazil or ARC-0704 strains were used to infect dishes at a ratio of 10 parasites to one host cell. After ON infection, cell cultures were washed with RPMI and incubated with 100 μM EdU for 24, 48 or 72 hr prior to fixation, staining and mounting in ProLong Diamond anti-fade mounting solution (ThermoFisher Scientific, Waltham, MA) containing 4',6-diamidino-2-phenylindole dihydrochloride (DAPI) to stain nuclear and kinetoplast DNA. Images were acquired using a laser scanning confocal microscope LSM 710 attached to an EXFO Xcite series 120Q lamp and a digital Zeiss XM10 camera.

## Ex vivo bioluminescence imaging

C57BL/6 mice were infected with $2.5 \times 10^5$ colombiana *T. cruzi* strain trypomastigotes co-expressing fluorescent (Tdtomato) and luminescent (luciferase) proteins. Sixty days post-infection, mice were injected i.p. with 10 mM EdU and sacrificed 12, 48 or 72 hr later. Before bioluminescence imaging, mice were transcardially perfused with 1x PBS and then with D-luciferin (0.3 mg/ml) diluted in 1x PBS. Tissue-specific ex vivo luciferase imaging was performed with tissues soaked in D-luciferin (0.3 mg/ml). Luciferase positive thick tissue sections were excised and consecutive imaging and sectioning were performed to reduce non-luminescent area and increase chances to localize amastigote infected tissues. Selected tissue sections were washed in 1x PBS and fixed with cold 4% PFA. After 4°C ON fixation, tissues were washed with 1x PBS and immersed in CUBIC clarifying solution (Susaki et al., 2015) (25 % N,N,N′,N′-Tetrakis (2-hydroxypropyl) ethylenediamine; 25% urea; 15% Triton X-100 and distilled water) with shaking at 37°C during 2 days. After clarification, EdU incorporation on intracellular amastigotes was detected as previously described.

## Tissue microscopy

For selection of parasite-containing tissue areas for further imaging in non-drug treated mice, luciferase-positive thick tissue sections were excised and consecutive imaging and sectioning were performed to obtain small tissue sections with the brightest luminescent foci. Tissue sections were fixed and clarified as described above, and EdU incorporation on intracellular amastigotes was detected by click chemistry as described above. Tissue sections were placed into sterile glass-bottom 35 mm petri dishes, mounted using ProLong Diamond anti-fade solution and prepared for confocal imaging using a Zeiss LSM 710 attached to an EXFO Xcite series 120Q lamp and a digital Zeiss XM10 camera. In the case of abdominal adipose tissue from drug-treated mice, wherein parasites were extremely scarce, both unfixed and PFA-fixed, non-clarified tissues were scanned for parasites using a 40X objective.

## DNA fragmentation detection using TUNEL assays

DNA fragmentation indicative of parasite death was assessed by TUNEL (Terminal deoxynucleotidyl transferase-mediated dUTP Nick End Labeling) assay according to manufacturer's protocol (Click-iT TUNEL Alexa Fluor 647 imaging assay; ThermoFisher Scientific, Waltham, MA). Briefly, Vero cells ($1 \times 10^5$) were infected with $1 \times 10^6$ colombiana strain trypomastigotes for approximately 12 hr. Seventy-two hours post-infection, cells were washed twice in PBS, fixed with 4% PFA and then washed again in PBS. After permeabilization with 0.2% Triton X-100, the cells were incubated with TdT reaction cocktail and washed twice with 3% BSA. Finally, cells were incubated with click-iT reaction buffer and washed with 3% BSA. Parasites were pre-treated for 10 min at room temperature with 10 IU/mL DNase I prior to the TUNEL for positive control. A negative control was performed in the absence of the terminal transferase. Cells were mounted using ProLong Diamond anti-fade mounting solution

containing DAPI for parasite nuclei visualization. Results were quantified counting about 200 cells in duplicate from three independent experiments.

## Labelling of parasites with fluorescent dyes

Cell suspensions of *T. cruzi* trypomastigotes were labelled with CellTrace Violet fluorescent dye (CellTrace Cell Proliferation Kit, ThermoFisher Scientific, Waltham, MA) and CFSE following manufacturer's instructions. Briefly, $2 \times 10^6$ trypomastigotes were incubated for 20 min at 37°C with 10 µM of CellTrace Violet or for 5 min with 5 µM of CFSE, protected from light. Unbound dye was quenched by the addition of five volumes of 10% FBS-RPMI or one volume of FBS respectively. After washing in fresh media parasites were used for infection of cell cultures or mice.

## Live cell imaging

Human foreskin fibroblasts (HFF) or Vero cells were seeded in 96 well glass bottom plates (Corning Life Sciences, NY) or 8 well 1 µ-slides (Ibidi, Fitchburg, WI) and infected with Tdtomato-expressing colombiana or CL strain parasites labeled with CellTrace Violet in ratios of 2:1 to 10:1 (parasites:host cells). For time-lapse video of amastigote replication, extracellular trypomastigotes were removed by washing and plates were placed in humid chamber with $CO_2$ and imaged at 40X magnification every 15 min for 48 hr in a Lionheart FX imager (BioTek, Winooski, VT). Images and time lapse videos were analyzed with the Gene5 software (BioTek). Live cell imaging of cultures at different times post-infection and after drug addition or removal were performed in a Cytation5 cell imager (BioTek) or a Delta Vision II Microscope System (GE Healthcare Biosciences, PA).

## Flow cytometry and cell sorting

Vero cell cultures were infected with CellTrace Violet-labeled Tdtomato-expressing parasites of colombiana or CL strain (5:1 ratio parasites:host cells) and incubated for 6 days at 37C in 5% $CO_2$. Free trypomastigotes released to the supernatant were harvested and imaged using an Image-Stream Mark II (MilliporeSigma, MA). CellTrace Violet- Tdtomato- positive parasites from those supernatants were sorted using a MoFlo XDP (Beckman Coulter, FL) and placed in fresh Vero cell cultures that were periodically monitored and imaged using a Cytation five imager (BioTek).

## In vitro drug treatment and washout assays

Gamma-irradiated Vero cells at 25,000/well were infected with 150,000 colombiana or CL strain Tdtomato-luciferase trypomastigotes in each well of a 96 well plate (Greiner Bio-one) for 24 hr. After removing non-infecting trypomastigotes, 10 µM BZN (approx. 10X $IC_{50}$) in complete RPMI media was added. The six replicate wells for each condition were read using BioTek Synergy Hybrid reader and images were taken by BioTek Cytation 5. At the indicated times post-treatment, BZN was removed, cultures washed and fresh medium without BZN added. Spent medium was removed and fresh medium with or without BZN were added weekly. Additional (15,000/well) Vero cells were added biweekly. The relative in vitro resistance to drug treatment by *T. cruzi* lines was determined as previously described (*Canavaci et al., 2010*), and IC50 was calculated using the GraphPad PRISM 5.0 software.

## Statistical analysis

The Mann-Whitney U tests and one-way variance analysis (ANOVA) of the GraphPad Prism version 5.0 software were used. Values are expressed as means ± standard error of mean of at least three separate experiments. P values equal to or minor that 0.05 were considered significant.

## Acknowledgements

We are grateful to Julie Nelson from the CTEGD Flow cytometry core and Muthugapatti Kandasamy from the Biomedical Microscopy Core as well as the assistance of Dr. Ronald Etheridge in live imaging. This work was supported by US National Institutes of Health grants AI108265 and AI124692 to RLT. Skillful technical assistance was provided by Katherine Kruckow.

## Additional information

### Funding

| Funder | Grant reference number | Author |
|---|---|---|
| National Institutes of Health | AI108265 | Rick L Tarleton |
| National Institutes of Health | AI124692 | Rick L Tarleton |
| National Institutes of Health | OD021719 | Rick L Tarleton |

The funders had no role in study design, data collection and interpretation, or the decision to submit the work for publication.

### Author contributions

Fernando J Sánchez-Valdéz, Conceptualization, Formal analysis, Investigation, Methodology, Writing—original draft; Angel Padilla, Conceptualization, Formal analysis, Supervision, Investigation, Methodology, Writing—original draft, Writing—review and editing; Wei Wang, Investigation, Methodology, Writing—original draft; Dylan Orr, Investigation, Methodology; Rick L Tarleton, Conceptualization, Supervision, Funding acquisition, Methodology, Writing—original draft, Project administration, Writing—review and editing

### Author ORCIDs

Rick L Tarleton (iD) http://orcid.org/0000-0002-9589-5243

### Ethics

Animal experimentation: This study was performed in strict accordance with the recommendations in the Guide for the Care and Use of Laboratory Animals of the National Institutes of Health. All of the animals were handled according to approved institutional animal care and use committee (IACUC) protocol A2015 05-010-R3 approved by the University of Georgia Institutional Animal Care and Use Committee under Animal Welfare Assurance #A3437-01.

### Decision letter and Author response

Decision letter https://doi.org/10.7554/eLife.34039.021
Author response https://doi.org/10.7554/eLife.34039.022

## Additional files

### Supplementary files

• Transparent reporting form
DOI: https://doi.org/10.7554/eLife.34039.019

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
