## [Decision Letter]

Thank you for submitting your article "Spontaneous dormancy protects *Trypanosoma cruzi* during extended drug exposure" for consideration by *eLife*. Your article has been reviewed by 3 peer reviewers, and the evaluation has been overseen by a Reviewing Editor and Wendy Garrett as the Senior Editor. The following individuals involved in review of your submission have agreed to reveal their identity: John M Kelly and David Sacks.

The reviewers have discussed the reviews with one another and the Reviewing Editor has drafted this decision to help you prepare a revised submission.

Summary:

In this study, the authors provide sound evidence that *Trypanosoma cruzi* can be found in a dormant/latent state both in vitro and in vivo experimental models. Using CellTRace Violet to track and isolate the few non-dividing parasites from infected cells in vitro, the authors were able to show that the dormant/quiescent parasites can establish a second round of infection in new host cells. They also provided good in vivo evidence that the non-dividing parasites persist following treatment with benzinidazole and are capable of re-initiating an infection. The data presented are novel and will be of much interest to those involved in finding curative treatments for this important neglected tropical infection. Indeed, the existence of non-replicating amastigotes is an important discovery that might explain the widely observed treatment failure as outcome of Chagas disease therapy and the poor efficacy of some compounds despite their promising curing properties.

Although no data on the mechanisms of this process are reported, the results are of significant value, and will no doubt stimulate much downstream research to unravel the mechanism(s), determine the biological role, and understand the consequences for drug development.

Essential revisions:

1) The in vitro infections shown in Figure 3 and 4 could have been exploited to provide more quantitative data regarding the proportion of non-dividing cells that arise within each generation, and to estimate if this frequency changes during successive generations as the numbers of intracellular amastigotes increases. It would also have been informative to more carefully quantify for comparison the subsequent infections initiated by the CellTrace Violet-positive and negative trypomastigote populations for their intracellular growth and the frequency of non-dividing forms that they give rise to. The limited data provided in Figure 5 shows a single Vero cell with a massive infection initiated by the CellTrace Violet-positive population. Only a single non-dividing parasite is indicated, which seems quite low compared to the frequencies observed in Figures 3 and 4.

2) The evidence that the non-dividing parasites are the only persisting organisms following treatment and that they are responsible for re-establishing infection is not so clear, particularly from the in vivo data:

The experiments shown in Figure 6 and Figure 6—figure supplements 1 and 2 are the key results of the paper but are difficult to assess. The results are meant to indicate that only non-dividing parasites remain after drug treatment, and that these parasites can reinitiate a productive infection. What is actually shown in Figure 6 from the tissue of 3-4 mice treated mice, however, is a single non-dividing parasite. It is not at all clear what this 'representative' result refers to. How many cells, or how much tissue was examined? Were there absolutely no Td-tomato staining cells that were CellTrace Violet-negative? Even a few may have contributed to the re-established infection. Is there some way to quantify this data, by flow cytometry perhaps? It would be especially informative to compare the number of the 'dormant' parasites in the adipose tissue (and/or other tissue) in groups of infected mice before and after treatment.

3) Overall the quality of the data presented needs to be improved:

Several fluorescence microscopy images are of poor quality, pixelated (e.g. Figure 3 24 hrs and 72 hr. Figure 4B, Figure 6B), saturated (e.g. Figure 2B, C; Figure 3 24 hrs and 72 hrs; Figure 2—figure supplements 1 and 2), out of focus (e.g. Figure 4F; Figure 5A Merge; Figure 5B; Figure 6B, C; Figure 7B); these images cannot be published as presented, considering the importance of the findings.

The use of CellTrace Violet to identify the dormant amastigotes is quite clever but some images of these parasites are rather poor (e.g. Figure 7B). By contrast, Figure 4E is of very good quality. Figure 4F, G: it would be nice to have images of samples from the other representative fractions of the cellTrace/Tc-Tomato plot.

4) Figure 7A: the authors assume that rebound fluorescence is due to dormant amastigotes:

No direct proof is presented since few resistant parasites (although rare) could give rise to the fluorescence peaks. Also, the authors should perform double staining with Tunel/EdU to confirm that dormant parasites are not dying. Biochemical evidence of the dormant amastigotes is lacking: can their ATP consumption be measured? Or the activity of critical metabolic enzyme(s)? Would it be possible to generate parasites transfected with fluorescence reporters for these components in order to directly monitor dormancy in vitro and in vivo? How do the authors distinguish dormancy from slow growth/long generation times?

---

## [Author Response]

1) The in vitro infections shown in Figure 3 and 4 could have been exploited to provide more quantitative data regarding the proportion of non-dividing cells that arise within each generation, and to estimate if this frequency changes during successive generations as the numbers of intracellular amastigotes increases. It would also have been informative to more carefully quantify for comparison the subsequent infections initiated by the CellTrace Violet-positive and negative trypomastigote populations for their intracellular growth and the frequency of non-dividing forms that they give rise to. The limited data provided in Figure 5 shows a single Vero cell with a massive infection initiated by the CellTrace Violet-positive population. Only a single non-dividing parasite is indicated, which seems quite low compared to the frequencies observed in Figures 3 and 4.

The reviewers ask interesting questions, but for the most part, they are not approachable with the current technology. For example, with neither the CellTrace Violet nor with EdU is it possible to assess if parasites stop dividing at later points (rounds) during infection in a host cell (i.e. once CellTrace Violet is diluted due to replication, it is difficult to observe cessation of replication without monitoring individual parasites by time lapse). We have done such time lapse monitoring of a limited number of host cells and have not yet observed amastigotes that were actively dividing, stop dividing. Further, in the in vivo and in vitro drug treatment/washout experiments, *all* parasites that are detected are CellTrace Violet-positive – and thus would have had to have arrested early in the replication cycle within host cells. Thus, we conclude that dormancy occurs preferentially and possibly exclusively soon after host cell invasion and conversion to amastigotes. It would be very challenging to generate highly quantitative frequency data on dormant parasites in different experiments (especially through 2 rounds of cell invasion as shown in Figure 5), under different culture conditions or with different isolates. Our attempts to do so suggest that there is not an increased frequency of dormancy in particular strains. With specific reference to the comment comparing the frequency of dormant parasites in host cells in Figures 3 and 4 compared to Figure 5, multiple trypomastigotes may enter the same host cell in our standard in vitro assays (where the standard parasite:host cell ratio is 10:1) while this is very highly unlikely in the experiment shown in Figure 5 where a very limited number of parasites were used to infect monolayers of host cells. The images we have selected for inclusion in this manuscript accurately reflect the fact that while dormancy occurs routinely, it is not uniform in every infected cell, hence our conclusion that it is not highly programmed.

2) The evidence that the non-dividing parasites are the only persisting organisms following treatment and that they are responsible for re-establishing infection is not so clear, particularly from the in vivo data:The experiments shown in Figure 6 and Figure 6—figure supplements 1 and 2 are the key results of the paper but are difficult to assess. The results are meant to indicate that only non-dividing parasites remain after drug treatment, and that these parasites can reinitiate a productive infection. What is actually shown in Figure 6 from the tissue of 3-4 mice treated mice, however, is a single non-dividing parasite. It is not at all clear what this 'representative' result refers to. How many cells, or how much tissue was examined? Were there absolutely no Td-tomato staining cells that were CellTrace Violet-negative? Even a few may have contributed to the re-established infection. Is there some way to quantify this data, by flow cytometry perhaps? It would be especially informative to compare the number of the 'dormant' parasites in the adipose tissue (and/or other tissue) in groups of infected mice before and after treatment.

We agree that the experiments in Figure 6 and Figure 6—figure supplement 1 and 2 are among the key results in the paper – although not the only ones. For these experiments, the peritoneal adipose tissue sample from each mouse is ~0.3 g of tissue spread over a surface of ~ 1.6 cm^2^ when mounted for microscopic analysis. In tissue harvested on day 6 of infection (prior to treatment), detection of infected cells is relatively easy and 3-4 infected cells can be observed within 15 minutes of scanning. Based on the observation of approximately 30 infected cells containing a total of 140 amastigotes, we estimate the frequency of Violet+ amastigotes among all amastigotes at this point is ~ 2%. Following 12 days of treatment this same amount of tissue must be exhaustively scanned for up to 3 hrs per sample to detect between 4 and 5 infected cells per sample. Every one of these infected cells contained only violet-positive parasite, generally only 1 parasite per cell. Scanning tissues from more than 12 mice using this process never revealed a violet-negative parasite during or at the conclusion of treatment. Figure 6 does indeed show a single dormant parasite – but the purpose of the supplementary figures is to indicate that this was not the only one found – collectively we have found dozens but did not feel that it was necessary to show them all. It is impossible to prove that there is *not* a non-dormant parasite in mice or animals after 20-30 days of drug treatment – but we have not ever observed them while we do, albeit with great effort, observe non-replicating parasites.

Some of this additional description has been added to the figure legends to indicate the vigor with which we searched for dormant and non-dormant parasites.

We do not believe that it will be especially informative to quantify dormant parasites before and after treatment in large part because the timing of “awakening” from dormancy is clearly variable. We know that brief BZN treatment of animals or cell cultures results in higher detectable levels of violet+ dormant parasites than longer-term treatment, suggesting that dormant parasites can exit dormancy while under drug treatment and as a result, be killed. One could argue that BZN is slowly killing parasites while they are dormant but then one would have to explain how some dormant parasites can resist BZN killing for 30 days or more.

Estimating the number of dormant parasites in the tissue by flow cytometry is quite difficult due to the scarcity of parasites surviving the BZN treatment and the process of the tissue necessary for flow cytometry (requiring enzymatic digestion, etc.). We unsuccessfully tried this approach originally with skeletal muscle of animals chronically infected with Td-tomato expressing parasites and moved to the microscopic analysis of adipose tissue with better results.

3) Overall the quality of the data presented needs to be improved:Several fluorescence microscopy images are of poor quality, pixelated (e.g. Figure 4B, Figure 6B), saturated (e.g. Figure 2B, C; Figure 3 24 hrs and 72 hrs; Figure 2—figure supplements 1 and 2), out of focus (e.g. Figure 4F; Figure 5A Merge; Figure 5B; Figure 6B, C; Figure 7B); these images cannot be published as presented, considering the importance of the findings.The use of Cell Trace Violet to identify the dormant amastigotes is quite clever but some images of these parasites are rather poor (e.g. Figure 7B). By contrast, Figure 4E is of very good quality. Figure 4F, G: it would be nice to have images of samples from the other representative fractions of the cellTrace/Tc-Tomato plot.

Overall quality of the images has been improved. An ImageStream example of Td-tomato positive- violet-negative parasite for Figure 4F is shown and sharper images selected for ImageStream figures. Images from live cultures using the Cytation 5 or from thick adipose tissue samples represent a challenge and usually those techniques have inherent limitations that prevent obtaining images of the same crispness as for those of monolayers or sections by the confocal microscopy (Figure 4E). We increased the resolution of all the pictures to 300dpi. In Figure 2B and C and Figure 7B the saturation and exposure were decreased as much as possible. In Figure 3, saturation and exposure were decreased in the pictures of 24hrs and 72 hrs. Pixelation and focus was corrected/optimized for Figure 4B, Figure 5Figure 6B and 7C. The image in Figure 6C at 6 days was changed to a picture containing a nest with a violet amastigote.

4) Figure 7A: the authors assume that rebound fluorescence is due to dormant amastigotes:No direct proof is presented since few resistant parasites (although rare) could give rise to the fluorescence peaks. Also, the authors should perform double staining with Tunel/EdU to confirm that dormant parasites are not dying. Biochemical evidence of the dormant amastigotes is lacking: can their ATP consumption be measured? Or the activity of critical metabolic enzyme(s)? Would it be possible to generate parasites transfected with fluorescence reporters for these components in order to directly monitor dormancy in vitro and in vivo? How do the authors distinguish dormancy from slow growth/long generation times?

The reviewers suggest a more detailed biochemical and metabolic characterization of the dormant parasites in order to discriminate them from proliferating amastigotes and to demonstrate true dormancy. We agree that this would be nice, but is well beyond this initial report. We have begun such investigations, attempting as suggested to engineer parasites expressing a reporter for ATP levels. However, although such reporters have been successfully used in mammalian cells (Tantama, et al., Nat Commun, 2013; Yaginuma, et al., Sci Rep, 2014), they did not function in *T. cruz*i, perhaps because ATP levels in the parasites are not within the reporter range of the cytoplasmic ATP sensor. Measuring enzymatic activity would also be useful, but very challenging in these intracellular parasites, where host enzyme activity and substrate availability must also be dealt with.

We unsuccessfully attempted a TUNEL/EdU double detection – the two protocols were incompatible in our hands. However, as described in Figure 3—figure supplement 1 the detection of TUNEL positive amastigotes dying inside cells is extremely rare and does not correlate with the proportion of EdU negative or violet-positive parasites detected in similar conditions. Further, we show that previously dormant parasites can convert to trypomastigotes, reinvade new host cells, and resume replication – all properties inconsistent with being dead.

Our experiments of infection of fresh cultures with sorted violet-positive parasites (Figure 5) as well as the rebound after drug treatment (Figure 6 and 7) suggest that parasites resistant to drugs under these conditions are not fixed indefinitely in a non-proliferative dormant state but are totally viable and capable of further proliferation. Therefore, it is not unreasonable to think that those parasites could just display a slower replication. Although we have observed in the time lapse experiments different frequency of division in some amastigotes, the lack of EdU incorporation in some parasites exposed to EdU in vitro for over nearly the entire replication cycle inside the cells, suggests that these parasites have not divided (or did so only just following invasion), in clear contrast to amastigotes actively proliferating in the same cell.

Parasites that resist BZN treatment for 20-30 days are not stably resistant to BZN in the classical sense- they can be killed as easily as parasites never exposed to BZN once they become metabolically active Figure 7—figure supplement 7). Thus, a few BZN-resistant parasites are not responsible for our observations, unless BZN-resistance is transient – e.g. the parasites are dormant. The difference between very slow growth and dormancy is ultimately semantics – if very slow growth means not dividing for 30 days, then we would agree with that description. But again, this is a transient property – once dormant parasites exit dormancy, they have a “normal” rate of growth. In short, we believe the sum of the data here argue for a state of dormancy. Determining the metabolic and biochemical markers of dormancy will not be easy but it is in our plans.